# Sheep Farmers’ Perception of Welfare and Pain Associated with Routine Husbandry Practices in Chile

**DOI:** 10.3390/ani8120225

**Published:** 2018-11-28

**Authors:** Cristian Larrondo, Hedie Bustamante, Carmen Gallo

**Affiliations:** 1Escuela de Graduados, Facultad de Ciencias Veterinarias, Universidad Austral de Chile, Valdivia 5090000, Chile; 2Instituto de Ciencias Clínicas Veterinarias, Facultad de Ciencias Veterinarias, Universidad Austral de Chile, Valdivia 5090000, Chile; hbustamante@uach.cl; 3Instituto de Ciencia Animal, Facultad de Ciencias Veterinarias, OIE Collaborating Centre for Animal Welfare and Livestock Production Systems—Chile, Universidad Austral de Chile, Valdivia 5090000, Chile; cgallo@uach.cl

**Keywords:** Animal welfare, husbandry practices, lambs, pain, sheep farmers, perception, agreement

## Abstract

**Simple Summary:**

Lambs are simultaneously subjected to several routine husbandry practices that cause pain. One of the main factors that limit the use of analgesics in lambs is the difficulty in pain recognition by sheep farmers. This study aimed to determine how husbandry practices are carried out in Chilean farms, the sheep farmers’ perception of animal welfare and pain, and the factors that affect them, as well as the level of agreement among farmers in the recognition of pain associated with these practices. Farmers were invited to participate in a workshop and they were asked through a survey about their sociodemographic information, how husbandry practices are being performed in their farms, and were asked to score the intensity of pain associated to seven of these practices. Castration and tail docking were perceived as the most painful practices and farmers agreed among them that these routine husbandry practices cause severe pain to animals. Several factors were associated with the farmers’ pain perception, such as the method used for the specific husbandry practices and the farmers’ educational level. In general, routine husbandry practices were carried out without using analgesics and with painful methods despite the agreement among farmers regarding the recognition of pain associated with these procedures.

**Abstract:**

Considering the public concern about the welfare of farm animals during routine husbandry practices, this study aimed to determine how husbandry practices are carried out in Chilean farms, sheep farmers’ perceptions of animal welfare and pain, and factors that affect them, as well as the level of agreement among farmers in the recognition of pain associated with these practices. Using a self-administered survey, participants were asked about their sociodemographic information, how husbandry practices are carried out in their farms, and their pain perception for seven of these common husbandry procedures using a numerical rating scale (0 to 10). A total of 165 farmers completed the survey and perceived castration and tail docking as the most painful practices in lambs (median pain score 10 vs. 8, *p* < 0.05). Pain perception was associated with the method used for the specific husbandry practices, the farmers’ educational level, the farm size, and flock size (*p* < 0.05). There was a fair to good level of agreement beyond chance (*p* < 0.05) in the recognition of pain associated with the most painful practices. In general, husbandry practices are not carried out in young animals, use painful methods, without using analgesics, which may have a negative impact on animal welfare.

## 1. Introduction

Pain is defined as a complex and subjective experience associated with actual or potential tissue damage [1], which not only depends on the severity of tissue damage, but also on the length of exposure of the animals to a painful stimulus [2]. Therefore, pain may potentially induce physical and behavioral changes that are the evidence of suboptimal animal welfare [2,3].

In production systems, sheep may experience pain due to different diseases, e.g., mastitis and lameness [4]. Also, lambs in extensive production systems, the predominant husbandry system in Chile [5], are subjected simultaneously to several routine husbandry practices that cause pain and distress (ear tagging, tail docking, castration, vaccination), which may have a negative impact on their welfare [6]. Different researchers [7,8] and international recommendations [9,10] challenge the use of some of these procedures in a routine way, such as castration and tail docking, because there is no consensus regarding the productive and welfare impact of these procedures on the animals [8,11,12]. Furthermore, society has become increasingly concerned about the welfare of farming animals, especially when they are subjected to painful husbandry practices [4,13]. Painful husbandry procedures in lambs induce an increase in locomotor activity, including abnormal postures, jumping, rolling, tail wagging, and repetitive standing and lying, regardless of the age and method with which they are performed [3,6,14,15].

Scientific evidence has shown that analgesics reduce pain during husbandry procedures [6,16,17]. The World Organisation for Animal Health recommends that when painful practices cannot be avoided, pain should be managed accordingly [18]. Similarly, the Chilean legislation [19] demands mitigation of animal suffering during painful husbandry practices and proposes four possible ways to reduce pain and improve animal welfare: (1) Replacing the current husbandry practice by another non-surgical procedure that has been demonstrated to improve animal welfare; (2) carrying out the husbandry practices at the earliest possible age; (3) use of analgesics during these procedures; or (4) genetic selection, selecting for animals that do not present the feature that requires the husbandry procedure. However, so far, there is still a low or lack of use of pain relief drugs in sheep farms when these procedures are performed [2,20]. Several reasons have been elicited for not using analgesics, including drug costs, availability, and practical difficulties associated with time intervals, herds’ sizes, and farm employees’ availability [4,21]. In addition to these limitations, both veterinarians and sheep farmers have reported difficulties in the recognition and evaluation of pain in lambs [4,22], a problem that further limits the operators’ motivation to reduce pain.

Pain evaluation and recognition in lambs is a complex process, mainly due to the fact that active pain behaviors in these animals are not as intense as in other species [4]. Moreover, pain recognition may be influenced by the incorrect assumption that younger animals have an immature central nervous system [23,24]. As prey animals, ruminants tend to be more stoic and not express pain, added to the impossibility in its verbalization [23]. Therefore, it is necessary that veterinarians and sheep farmers know how to correctly recognize pain in these species, and implement pain management strategies when husbandry practices are carried out. The aims of this study were to determine how husbandry practices are carried out in Chilean farms, to describe sheep farmers’ perception of animal welfare and pain, the factors that affect them, as well as the level of agreement among farmers in the recognition of pain associated with these practices.

## 2. Materials and Methods

This study was approved by the Committee of Bioethics and Use of Animals in Research of the Universidad Austral de Chile (Nº 288/217). The study was conducted between November 2016 and May 2018. A self-administered survey was applied to sheep farmers based in three Chilean regions: Magallanes and the Chilean Antartic (sheep population: 1,571,056; 77.1% of the national population), Aysén (177,972; 8.7%), and Libertador Bernardo O’Higgins (123,715; 6.07%). These regions were selected because they have the largest number of sheep and flocks of the country [25]. Through government and private agencies, sheep farmers in each region were invited to participate in a workshop entitled, “Improving handling practices on sheep farms”, where veterinarian experts spoke about sheep production, livestock practices, and animal welfare. Before each workshop started, a self-administered, confidential, and anonymous survey was given to participants to be completed. Of a total of 747 existing farmers in the mentioned regions [25], 180 attended the workshops and took the survey, and 165 completed all questions. However, in the case of the numerical rating scale for pain perception of routine husbandry practices, only 125 farmers rated all husbandry practices. The data obtained represent a sample size with a confidence level of 95%, with an error equal to 7% (R Core Team, Vienna, Austria) [26], and correspond to 22% of the sheep farmers in the above-mentioned regions.

### 2.1. General Sociodemographic and Farm Information of Sheep Farmers

The survey consisted of three sections. In the first section, open-ended questions about the farmers’ sociodemographic information were obtained, including gender, age, region, educational level, as well as information related to their time experience as a sheep farmer (years), farm size (hectares), and flock size (all sheep). Also, farmers were asked to select one or more options regarding if their farm had technical advice in the following areas: Nutritional, reproductive, health, or none of them.

### 2.2. Routine Husbandry Practices

The second section of the survey contained open-ended questions of how sheep farmers carry out the following husbandry practices in lambs: Animal identification, tail docking, and castration. For each procedure, they were asked by open questions if they carry out each practice, and if they do, “at what age they carry it out”, the method used, the reasons for performing the practice, person responsible for performing the procedure, and if they use analgesics or not. Additionally, for tail docking, a diagram was used to mark the length at which the tail was docked and to justify the answer. The diagram was adapted from Fisher et al. [8], and farmers had to mark one of three options for tail length: Short = docked leaving little tail, not enough to cover the tip of the vulva in ewe lambs and at similar length in male lambs; Medium = tail stump covers the tip of the vulva in ewe lambs and at similar length in male lambs; and Long = docked longer than Medium.

### 2.3. General Animal Welfare Perception

The third section aimed to establish the farmer’s knowledge concerning animal welfare, including individual perception and importance. Farmers were asked if they had previously heard about animal welfare (yes/no), and to choose what animal welfare means to them by selecting one or more of the following options: Animals do not suffer from hunger and thirst; animals do not suffer from pain, injury, and diseases; animals do not suffer from discomfort; animals do not suffer from fear and distress; animals can express normal behavior. Participants were also asked to select one or more options from a pre-defined list regarding the importance of having good animal welfare in their farms, considering the following alternatives: It affects animal health and production; it influences meat quality; it is a consumer requirement; or animal welfare is indifferent to me. Also, they were asked to select and rank in order of importance the three main animal welfare issues in their farms from a pre-defined list: Predation, lamb mortality, rounding up/transportation, husbandry practices, slaughter, facilities, nutrition/water availability, and diseases.

### 2.4. Farmers’ Perceptions of Pain during Routine Husbandry Procedures

Finally, sheep farmers were asked to describe how they perceive the intensity of pain associated with each of the following husbandry procedures: Animal identification, tail docking, castration, hoof trimming, deworming, vaccinating, and shearing. For this, a table with an eleven-point (0 to 10) numerical rating scale (NRS) was used, considering 0 as “no pain’’ and 10 as the maximum pain [26].

### 2.5. Data Management and Statistical Analyses

Data of a total of 165 completed surveys were analyzed using the statistical program, R (R Core Team, Vienna, Austria) [27]. Descriptive analysis of sociodemographic information, husbandry procedures, and the animal welfare section were performed. Sociodemographic information and open-ended responses from the second and third sections (routine husbandry procedures and general animal welfare perception) were coded as frequencies and percentages. The scores obtained in the NRS for each husbandry practice were summarized with their mode, median, and range. Also, a radar chart was made with the median pain scores of each husbandry practice. Spearman correlations were made between pain scores (median pain scores) that sheep farmers perceived as being associated with husbandry practices. Mann-Whitney non-parametric tests (*p* < 0.05) were used to determine the association of the method used for each husbandry practice with the sheep farmers’ pain perception. Linear regression models with Poisson distribution were fitted to assess the effect of the continuous variables (age, time experience as a sheep farmer) and factors (gender, educational level, flock size, farm size) on the pain scores associated to each husbandry practice.

To estimate the degree of agreement among sheep farmers in the recognition of pain associated to husbandry practices, pain scores of NRS were categorized as follows: No pain = 0; Mild to moderate pain = 1 to 5; and Severe pain = 6 to 10. The degree of agreement beyond chance was estimated by the Fleiss’ kappa coefficient (κ) for multiple raters using Epidat 4.1 software, where κ values: <0.40 indicated poor agreement; between 0.40 and 0.75 fair to good agreement; >0.75 indicated excellent agreement [28].

## 3. Results

### 3.1. General Sociodemographic and Farm Information of Sheep Farmers

The majority of sheep farmers were males (*n* = 145; 87.9%), 18 (10.9%) were females, and only two persons did not answer this question (1.2%). The mean age of the participants was 47.3 ± 14.5 years, the mean time experience as sheep farmer was 23.5 ± 19.9 years. More than half of the farms had technical advice, either from the government (63/165; 38.2%) or private (41/165; 24.8%). In addition, most of the farms (115/165; 69.7%) were visited by a veterinarian, at least once a year. Table 1 shows further information about the age of farmers, their educational level, number of ewes per farmer (flock size), and farm size (ha).

### 3.2. Routine Husbandry Practices

#### 3.2.1. Identification

Animal identification was carried out (85.1%) on lambs at a mean age of 3.8 ± 3.1 months (range= 0.5–18). The most common method for identification was ear notching (52.1%), followed by the use of an ear tag (39.4%), and both methods (8.5%). The totality of farmers (100%) considered identifying their animals as a sign of property and because the procedure allowed them to maintain accurate records. No farmers reported the use of analgesics (100%) during this procedure, which is mainly performed by themselves (33.9%) or by farm personnel (26.3%). Nonetheless, 33% of farmers did not answer the question of who performed the procedure.

#### 3.2.2. Tail Docking

Of the sheep farmers, 91.2% answered that they dock lamb tails at a mean age of 3.4 ± 1.9 months (range = 1–12). The most common method reported was the use of a knife (67.3%), followed by a rubber ring (18.7%), and hot iron (11.2%). Reasons for tail docking included: To improve mating (29.1%), to improve animal sanitary conditions (22.3%), for easier handling (21.4%), esthetic reasons (10.7%), and to improve reproduction (6.8%); 3.9% of farmers answered that they do not know why they do it. Other less frequent answers (< 2%) included: Tradition, to improve animal productivity, and as a method of animal identification. The majority of sheep farmers (85.9%) do not use analgesics during this husbandry practice. Similar to animal identification, tail docking is performed by sheep farmers (40.6%) and by farm personnel (29.7%).

The majority of sheep farmers (49.7%) performed tail docking only in animals that will remain in the farm for breeding purposes, 25.9% only dock ewe lambs and 10.9% perform it on male and ewe lambs. Most farmers (55.9%) leave a short tail stump that does not cover the tip of the vulva in ewe lambs and at a similar length in male lambs; 33.6% dock lambs’ tails at a medium length and 0.8% at a long length. The main reasons given by farmers in relation to why they dock lambs’ tails at these lengths were: To improve mating (33.8%), to improve animal sanitary conditions (18.4%), tradition (12.2%), esthetic reasons (10.2%), and to protect the perineum (5.4%).

#### 3.2.3. Castration

Regarding castration, 52.3% of the sheep farmers castrate their male lambs at a mean age of 7.6 ± 15 months (range = 1–72). The most common method used was a rubber ring (75.5%), 14% use a knife, and 10.5% use other methods. The main reasons for castrating their lambs included: Handling (48%), commercial requirements (20%), improvement of animal productivity (17%), and to avoid breeding (15%). Most of the farmers (92.6%) indicated that they do not use analgesics during castration with the procedure mainly being performed by farm personnel (40.8%) or by themselves (32.4%).

### 3.3. General Animal Welfare Perception

The majority of sheep farmers (83.6%; 138/165) had previously heard about animal welfare. Regarding animal welfare [29], 83.6% of participants selected all the options given, as explained in Section 2.3, meaning that animals do not suffer from thirst, hunger and malnutrition, pain, injuries, diseases, discomfort, fear, and distress, and the possibility to express normal behaviors. Five percent of farmers perceive and associate animal welfare only with good nutrition and 3.6% associate it only with the fact that animals may express their natural behavior.

The importance that sheep farmers give to animal welfare was mainly associated with the facts that it affects animal health and production (53.55%), it influences meat quality (28.9%), and it is a consumer requirement (12%). A 5.6% of participants answered that animal welfare was something indifferent to them. The most important animal welfare concerns reported by sheep farmers on their farms, in order of importance, included: Nutrition and water availability > diseases > predators. In general, sheep farmers perceived that animal welfare in their farms was good (65.7%) and regular (32.8%), with a minority perceiving that it was bad (1.5%).

### 3.4. Sheep Farmers’ Perceptions of Pain during Routine Husbandry Procedures

A total of 125 answers were obtained for the pain NRS, which represents a response rate of 75.8% (125/165). Only data from farmers that completed all the NRS (seven husbandry practices) was included. Husbandry practices that farmers scored as the most painful procedures included both castration and tail docking (Table 2). Castration was perceived to be more painful than tail docking (median pain score 10 vs. 8; *p* < 0.05). The most frequent pain scores for castration and tail docking were 10 (mode 10), with 55.2% and 35.2% of sheep farmers scoring both procedures with a score of 10, respectively (Table 2). As shown in Figure 1, animal identification was perceived as the third most painful husbandry practice followed by vaccination (median pain score 4 vs. 2; *p* < 0.05). Hoof trimming, shearing, and deworming were scored with the lowest pain scores in the NRS (Figure 1).

A positive and significant correlation was obtained (rho = 0.48; *p* < 0.05) between castration and tail docking pain scores and between tail docking and identification pain scores (rho = 0.21; *p* < 0.05), regardless of the method used for these procedures. In contrast, castration and identification pain scores were not correlated (rho = 0.07; *p* > 0.05). Tail docking and animal identification using a knife were scored with higher scores (*p* < 0.05) by sheep farmers, compared to when these procedures were performed using a rubber ring and ear tagging, respectively (Table 3). The farmers did not perceive differences in pain (*p* > 0.05) between tail docking with a knife and hot iron, nor between castration methods (*p* > 0.05).

The age of farmers affected (*p* < 0.05) their pain perception only for deworming and vaccinating (Table 4). There was a significant effect (*p* < 0.05) of farmers’ educational level on pain perception associated with the majority of husbandry practices (Table 4). Pain scores were higher in farmers who did not complete a formal education curriculum and who had elementary education compared to those with professional education (*p* < 0.05). The time experience as sheep farmers (*p* < 0.05) only affected the farmers’ pain perception for vaccinating. Flock size and farm size were factors that affected (*p* < 0.05) farmers’ pain perception associated with hoof trimming, deworming, and vaccinating. Farmers who had larger flocks and farms gave lower pain scores (*p* < 0.05) for these husbandry practices than farmers who had smaller flocks and farms (Table 4).

Overall, agreement beyond chance among sheep farmers for pain recognition associated to husbandry procedures was poor (Fleiss’ κ = 0.3353, *p* < 0.05; Table 5). Similarly, agreement for pain recognition associated with the “No pain” (κ = 0.1896) and “Mild to moderate pain” (κ = 0.2311) categories was also poor (*p* < 0.05). Nonetheless, for the “Severe pain” category, the agreement among farmers was fair to good (κ = 0.5603, *p* < 0.05; Table 5).

## 4. Discussion

### 4.1. Sheep Farmers and Routine Husbandry Practices’ Description

Sociodemographic features of sheep farmers (Table 1) are similar to the current national situation, in which the majority of farmers are males [25]. The average age of sheep farmers is in agreement with the increased aging tendency of Chilean farmers and the general population [25,30] and also corresponds with the experience as sheep farmers, which in this study exceeded 20 years. The educational level here reported for farmers is in agreement with another Chilean study [30], in which an average of 6.2 years of education with incomplete secondary education is described. Additionally, in the present study, the majority of sheep farmers (38.7%) had completed elementary education and more than half (51%) had completed secondary education (Table 1).

A marked heterogeneity in flock size is described, characterized by a similar percentage of sheep farmers with less than 50 (36.4%) and more than 500 ewes (30.3%). Sheep farmers in the Chilean Patagonia (Regions of Magallanes and Aysén) have large flocks (more than 500 ewes) in extensive production systems and large farms (more than 1,000 hectares). In contrast, sheep farmers of central Chile (Region of Libertador Bernardo O’Higgins) are characterized by small flocks (less than 50 ewes) and less than 50 hectares. These differences between regions in terms of flock and farm size are in agreement with Gallo et al. [5], and further relate to differences in transport and preslaughter conditions of the lambs produced.

Current European and Chilean legislation [9,10,19] mandate performing painful husbandry practices in young animals. However, the results of the present study indicate that painful husbandry practices in lambs, such as animal identification, tail docking, and castration, are not performed accordingly. The age at which sheep farmers perform these procedures could be associated with the predominant sheep production conditions in Chile, where lambs are reared in large herds (thousands of animals) and extensive grass pastures [5]. These factors make the rounding up and gathering of animals in their early weeks of life quite difficult. Consequently, the vast majority of routine husbandry practices are performed simultaneously [31], usually at ages older than three months, which may have negative implications on animal welfare [20,21,23,32]. Moreover, tail docking at 45 days may induce long-term consequences, such as primary hyperalgesia and chronic pain [33].

The most frequent method used by sheep farmers to carry out identification and tail docking in lambs was the knife. Several authors [16,34,35] have identified the knife as the most painful method for both ear notching and tail docking, resulting in greater behavioral and physiological changes than ear tagging and using a hot iron, respectively [35]. Therefore, current scientific evidence recommends avoiding the use of a knife, selecting instead methods that are less painful [16,34,35,36]. Similarly, rubber ring castration was preferred by Chilean sheep farmers, probably due to the lower costs implied, easiness to perform, and relative quickness. However, some studies suggest that rubber ring castration causes greater pain when associated with behavioral changes compared to other castration methods [6,23,37].

Animal identification and castration are performed mainly for handling and commercial reasons, while there are several reasons why sheep farmers carry out tail docking, including improving both mating and farm sanitary conditions. According to the scientific evidence, these arguments are questionable because we are not aware of any studies reporting that tail docking improves mating [38,39]. Moreover, according to Orihuela et al. [39], rams prefer to court and mate with intact tail ewes over tail docked ewes. Also, there is no consensus in the scientific literature about a potential relationship between dags/flystrike and tail presence. Fecal soiling and dags (fecal material around the anus) could be mainly associated with fecal consistency [12,40], so it is essential that farms have proper management practices, such as deworming, shearing, and crutching. Furthermore, in countries where flystrike is not a welfare and health issue, like Chile, tail docking may be more difficult to justify as a routine husbandry practice [10].

Sheep farmers mostly performed tail docking to ewe lambs and to those ewe and male lambs that remain in the farms for breeding purposes. This fact could also be associated with the main reason why they perform this practice: To improve mating. Similarly, the length at which farmers tail docked their animals may be also influenced by this factor, mainly due to the fact that the vast majority of sheep farmers carry out this procedure, leaving a short tail that does not cover the tip of the vulva. However, it has been demonstrated that short-tail docked animals have a greater incidence of rectal prolapses and even a higher risk of flystrike [8,41]. Therefore, several authors [8,41,42] have recommended that when tail docking is performed, a tail stump that covers the tip of the vulva in ewe lambs with a similar length in male lambs should be maintained.

When painful husbandry practices cannot be performed at an early age, legislation mandates for the use of analgesics [19,20]. This is supported by the fact that an increased chronic inflammatory reaction has been demonstrated in lambs that have been castrated at later ages in contrast to when these practices are performed in younger lambs [43]. These differences could be associated with an increased tissue trauma due to the procedure in older animals, rather than differences in pain sensitivity related to age or an underdeveloped central nervous system [4,20,23]. According to the present study, nearly all farmers carry out husbandry practices in lambs without the use of pain relief drugs. These results are in agreement with other international results, where analgesic drugs are not routinely administered during husbandry practices, despite the fact that international legislation from some countries requires or promote their use [9,10]. There are many reasons why farmers do not administer analgesics to lambs, including management and economical arguments [44]. Nevertheless, analgesics’ use during and after various husbandry practices have shown to significantly improve animal welfare [16,17]. Accordingly, Small et al. [17] administered meloxicam before tail docking and reported a seven-fold reduction in abnormal pain associated behaviors. Furthermore, Phillips et al. [45] mentioned that Australian sheep farmers considered analgesic administration more important than the method used to carry out castration and tail docking.

### 4.2. Sheep Farmers’ Perceptions of Animal Welfare and Pain

The majority of sheep farmers have previously heard about animal welfare and they associate it to physical, mental, and behavioral conditions of animals. Also, more than half of the surveyed farmers agreed that animal welfare is important to ensure animal health and production. These findings are in agreement with those reported by Australian [46] and Brazilian farmers [47]. The most important animal welfare issues identified by sheep farmers were associated with nutrition and water availability. Interestingly, almost all surveyed farmers perceived that animal welfare on their farms was “regular to good”, which, according to them, would be associated to the extensive production conditions, in which animals are able to express their natural behavior and human handling is infrequent [21,48,49]. However, animal welfare in extensive production systems is affected by several environmental challenges and conditions, such as drought, snow, food and water scarcity, and predation. Moreover, the close observation of animals by a stock person or farmers is difficult [21,48,49]. Consequently, animal welfare perception in extensive production systems may be overestimated in contrast to intensive production systems [48].

Castration and tail docking were the only husbandry practices in which the totality of sheep farmers agreed as inducing pain (Table 2). These results are contrary to those reported by Dwyer [21] and Tamioso et al. [47] in the United Kingdom and Brazil, respectively. In the study of Dwyer [21], 15.8% of farmers indicated that tail docking was a painless procedure and of the remaining 84.2% only associated this practice with mild to moderate pain (range 2–6, in a 1 to 10 scale). The results of Dwyer [21] and Tamioso et al. [47] reported that castration was associated with mild to severe pain. However, Dwyer [21] found a positive correlation between tail docking and castration scores, similar to the findings obtained in the present study.

Castration was perceived by sheep farmers as the most painful husbandry practice for lambs, followed by tail docking and animal identification (Table 2), which is in agreement with some scientific evidence [6,26,47]. Using a similar pain scale, Scott et al. [26] also found that veterinarians gave higher pain scores to rubber ring castration than tail docking (median pain score 6 vs. 4). These results are different from those obtained in the present study, where the sheep farmers’ pain perception was higher for castration and tail docking regardless of the method used (median pain score 10 vs. 8).

The pain perception differences found between sheep farmers in relation to the method used to perform certain husbandry practices (Table 3) are in agreement with those reported in other pain perception studies [26,45] and animal behavior research [6,16,50]. Routine husbandry practices, such as animal identification, tail docking, and castration carried out using a knife, were perceived by sheep farmers as more painful. However, the use of a knife was the preferred method used by sheep farmers to perform the vast majority of painful husbandry practices in lambs, which does not agree with their pain perception. It is important to highlight that sheep farmers’ perception could be based only on acute pain experienced by animals, and chronic pain associated with some husbandry practices, e.g., tail docking, is invisible to sheep farmers [33].

Flock size and farm size influenced the perception of pain (Table 4). Sheep farmers with large flock and farm sizes perceived less pain associated to husbandry practices than those farmers with smaller flock and farm sizes, presumably because animals in extensive production conditions are infrequently gathered and observed, and are subjected at the same time to the vast majority of painful practices [31]. When flock size is large, husbandry practices must be performed faster, thus limiting the observation of any pain related behaviors, which are mostly acute and occur during the first 30 to 60 minutes after painful procedures [37,50]. Additionally, one of the actual trends in extensive sheep production systems is to decrease the stockperson:sheep ratio [21,51]. Therefore, sheep farmers’ perceptions of animal welfare and pain associated to husbandry practices in large flocks may be deficient, mainly to a decreased human:animal interaction, a situation that could be different for farmers with small flocks. Managers of larger flocks may be distanced from participating in painful routine procedures.

The majority of sheep farmers perceived hoof trimming, deworming, vaccinating, and shearing as painless practices, although a high variability in their pain perception was observed (Table 2). This variability could be explained by the effect of sociodemographic factors on pain perception associated with these husbandry practices (Table 4). According to the results of our study, farmers who had a higher educational level had a lower pain perception; these farmers also had larger flock sizes. Having larger flocks may reduce the human:animal interaction and also pain perception because they do not see and handle their animals frequently; on the contrary, farmers with smaller flocks may have a closer human-animal bond and hence a higher perception of pain in their animals because they round them up every day and have greater chances of watching them closer. Additionally, this variability could be associated to the fact that these husbandry procedures have the potential to induce pain when they are carried out improperly, e.g., when the shearer cuts the sheep’s skin [52]. These husbandry practices could be more directly related to handling procedures, such as the previous rounding up, confinement, food and water deprivation, and isolation from the flock [53,54]. Therefore, it is important to train stockmen and sheep farmers to handle animals carefully and carry out these routine practices properly to improve animal welfare.

The poor degree of agreement observed among sheep farmers for pain recognition associated to husbandry practices (Table 5) could be also explained by difficulties in animal pain assessment [4,22,48,55] rather than farmers being insensible to the pain animals may experience [46]. Pain assessment difficulties in ruminants have been described as one of the most important reasons why veterinarians and farmers do not administer analgesics when they carry out painful husbandry practices in sheep [4,22,55]. For these reasons, several researchers have developed guidelines and methodologies to more accurately assess pain in sheep [22,50]. Another possible explanation of the poor agreement among sheep farmers is the variability of the scoring obtained mainly for routine husbandry practices that should not cause pain to animals, such as deworming, vaccination, shearing, and hoof trimming (Table 2). However, for the “severe pain” category, the level of agreement among sheep farmers was fair to good, indicating that there is an agreement in the recognition of intense pain associated to husbandry practices causing mild to severe pain to animals, such as castration and tail docking.

This study may be limited by the selection of farmers that attended an industry-government continuing education event and a self-administered survey format. This process may have selected from a cohort of more generally literate farmers than the population targeted. This is the first study in Chile that approaches how sheep husbandry practices are performed and the farmers’ perception of pain and animal welfare; further studies on the subject should follow.

## 5. Conclusions

Animal identification, tail docking, and castration are painful husbandry practices that are carried out by Chilean sheep farmers at later ages than recommended by the international literature, using methods that may have a negative impact on animal welfare and analgesia is rarely used. Sheep farmers perceived castration and tail docking as the most painful husbandry practices, however, they carried out these procedures with methods that they perceived as the most painful for animals. The vast majority of sheep farmers perceived a regular to good level of animal welfare in their farms, possibly due to the fact that they considered other animal welfare issues as more important than husbandry practices, such as animal nutrition and water availability. Although there was a poor global agreement among sheep farmers in the recognition of pain associated to routine husbandry practices, there was a good level of agreement for the most painful husbandry practices, such as castration and tail docking, notwithstanding that no analgesia is used for both procedures. It is inferred that the lack of use of analgesics would not be explained by sheep farmers’ pain perceptions nor their time experience as sheep farmers, but other important factors that influence this decision would exist, such as practical and economic reasons, as well as the fact that in extensive pasture production systems, sheep farmers do not have a close human:animal interaction and there are difficulties to actually observe animals.

## Figures and Tables

**Figure 1 animals-08-00225-f001:**
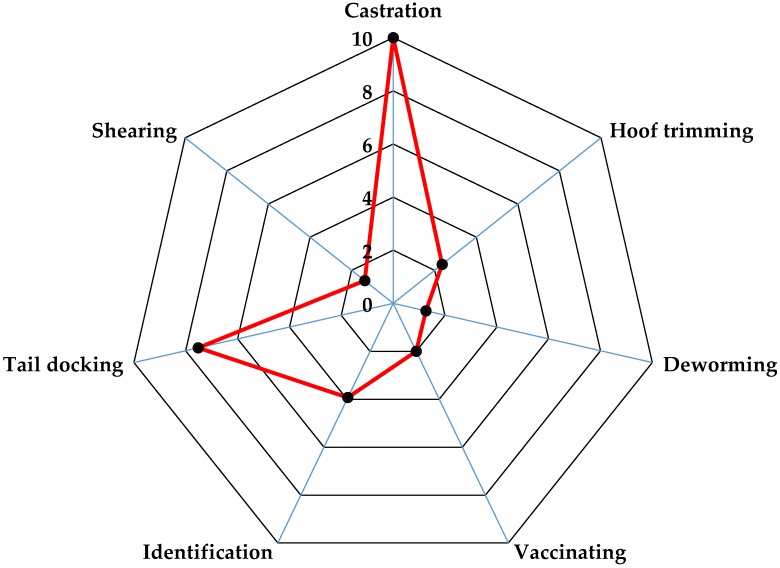
Median pain scores associated to husbandry practices in lambs using a numerical rating scale, from 0 (no pain) to 10 (maximum pain).

**Table 1 animals-08-00225-t001:** Sociodemographic data of sheep farmers (*n* = 165) and farm information.

Variable	Frequency	Percentage
Age (years)		
18–32	25	15.8
33–47	64	38.8
48–62	48	29.1
≥ 63	26	15.8
Educational level		
Elementary ^1^	57	34.5
Secondary ^2^	34	20.6
Technical	21	12.7
Professional	37	22.4
No schooling	7	4.2
Flock size		
≤ 50	60	36.4
51–200	33	20
201–500	10	6.1
501–1000	6	3.6
≥ 1001	44	26.7
Farm size (ha)		
≤ 50	57	34.5
51–1000	40	24.2
1001–5000	13	7.9
5001–10,000	12	7.3
≥10,001	13	7.9

^1^ Eight years of elementary education; ^2^ four years of secondary education.

**Table 2 animals-08-00225-t002:** Summary statistics of sheep farmers’ (*n* = 125) perceptions of pain intensity associated to seven husbandry practices in lambs using a numerical rating scale, from 0 (no pain) to 10 (maximum pain).

Husbandry Practice	Mode	Median	Range
Castration	10	10	2–10
Hoof trimming	0	2	0–10
Deworming	0	1	0–7
Vaccinating	2	2	0–10
Identification	3	4	0–10
Tail docking	10	8	2–10
Shearing	0	1	0–6

**Table 3 animals-08-00225-t003:** Median numerical rating scale pain scores perceived by sheep farmers according to husbandry practice and method used.

Husbandry Practice	Method			
	Knife	Ear Tagging	Rubber Ring	*p*-Value
Identification	5	3	-	*p* < 0.05
Tail docking	9	-	5–6	*p* < 0.05
Castration	10	-	8–9	*p* > 0.05

Numerical rating scale, from 0 (no pain) to 10 (maximum pain).

**Table 4 animals-08-00225-t004:** Effect of sociodemographic factors on pain perception associated with routine husbandry practices in lambs. Results of the linear regression models testing pain scores for each husbandry practice.

Main Effect	Husbandry Practice						
	Castration	Hoof Trimming	Deworming	Vaccinating	Identification	Tail Docking	Shearing
Gender	ns	ns	ns	ns	ns	ns	ns
Age (years)	ns	ns	*p* < 0.05	*p* < 0.05	ns	ns	ns
Education	ns	*p* < 0.05	*p* < 0.05	*p* < 0.05	*p* < 0.05	*p* < 0.05	ns
Experience	ns	ns	ns	*p* < 0.05	ns	ns	ns
Flock size	ns	*p* < 0.05	*p* < 0.05	*p* < 0.05	ns	ns	ns
Farm size	ns	*p* < 0.05	*p* < 0.05	*p* < 0.05	ns	ns	ns

* ns: Not significant effect (*p* > 0.05).

**Table 5 animals-08-00225-t005:** Level of agreement among sheep farmers (*n* = 125) in the recognition of pain associated with husbandry practices in lambs. Fleiss’ kappa coefficient values for pain categories: No pain (0), Mild to moderate pain (1–5), Severe pain (6–10), according to pain intensity scores used in a numerical rating scale.

Category	Kappa	Confidence Interval (95%)		Z-Value	*p*-Value
No pain	0.1896	0.0572	0.3163	44.1683	*p* < 0.05
Mild to moderate pain	0.2311	−0.0384	0.4894	53.8356	*p* < 0.05
Severe pain	0.5603	0.2204	0.8776	130.4994	*p* < 0.05
Global kappa	0.3353	0.0507	0.6049	105.6649	*p* < 0.05

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
