# Peer review of "Sheep Farmers’ Perception of Welfare and Pain Associated with Routine Husbandry Practices in Chile"

_animals, 2018, doi:10.3390/ani8120225_

Round 1

Reviewer 1 Report

Most comments are on clarity of language and minor comments on the statistical presentation. I find the reporting of power of the study line 98 of the submission an commendable and often neglected component of this type of study.

I do not know the details of the statistical program used but assumed they were calculating the "agreement beyond chance" and the Fleiss' kappa statistic and encourage the authors to make that adjustment.

There was no mention of the stress associated with handling which was intentional .  Stress in this area of research is a meaningless word and it is clear that this paper was on handler perception of pain in animals.  If they could somehow remove the word "stress" from line 388 and 391 I would consider it an improvement on the clarity of the discussion.

Table 4 is reporting on the regression model.  The table is located between table 3 which appears to be an agreement beyond chance report and table 5 which also appears to be an agreement beyond chance report.  

I would suggest switching the location of table 4 and 5 in the document and perhaps direct the readers attention to the new method of analysis for the data in current Table 4.

I converted the original manuscript to Word made some suggestions in the document and saved it back into a .pdf for some English language clarification suggestions. I think I have attached that draft using the button below if not notify me by e-mail

Author Response

We appreciate the reviewer’s comments and suggestions. We have considered almost all these recommendations in order to improve our paper.  

Response to Reviewer 1 Comments

Most comments are on clarity of language and minor comments on the statistical presentation. I find the reporting of power of the study line 98 of the submission an commendable and often neglected component of this type of study.

Action: We have introduced all these changes in the revised document. We used the pwr library from Rstudio statistical program, line 98

I do not know the details of the statistical program used but assumed they were calculating the "agreement beyond chance" and the Fleiss' kappa statistic and encourage the authors to make that adjustment.

Action: We agree with reviewer comment and we have added this information, see line 159-160.

There was no mention of the stress associated with handling which was intentional .  Stress in this area of research is a meaningless word and it is clear that this paper was on handler perception of pain in animals.  If they could somehow remove the word "stress" from line 388 and 391 I would consider it an improvement on the clarity of the discussion.

Action: It has been removed, also in line 414

Table 4 is reporting on the regression model.  The table is located between table 3 which appears to be an agreement beyond chance report and table 5 which also appears to be an agreement beyond chance report. I would suggest switching the location of table 4 and 5 in the document and perhaps direct the readers attention to the new method of analysis for the data in current Table 4.

Action: We understand your suggestion. However, Table 3 is not an agreement beyond chance report. Table 3 shows differences on farmer’s pain perception according to husbandry practice and method used. These results were obtained using the median pain scores that farmers assigned to each husbandry practice when they carried out the procedure using the method that they reported in the survey. Hence we did not switch the tables.

I converted the original manuscript to Word made some suggestions in the document and saved it back into a .pdf for some English language clarification suggestions. I think I have attached that draft using the button below if not notify me by e-mail

Action: We have introduced all these changes in the revised document.

Reviewer 2 Report

I am very interested in this study however there are two major problems I see in addition to what is listed below: 1) The paper needs extensive editing for grammar and 2) I would like to see the survey instrument given to farmers (as well as an information on how responses were coded)

Line 14 - This sentence is very long. Please reword to be more precise

Line 18 - Please mention how the farmers were recruited

Line 22 - Grammar: practices not practice. Should be farmer’s not farmers

Line 23 - I fail to see how the authors judgment that painful practices have negative impact on welfare is relevant to this study. I would also avoid the subsequent judgment that farmers don’t do things “properly”. Properly according to whom?

Line 43 - ungrammatical. Also please consider using the ISAP definition which stipulates that tissue damage is neither necessary or sufficient for pain.

Line 45 - why do you define pain twice?  What is the point?

Line 48 - diseases is clear, but conditions is very vague consider removing or clarifying

Line 52 - It is not clear to me what this line is saying.  Researchers disagree about what? I doubt that they disagree that these procedures are painful, but rather about what ought to be done about the pain or how important it is.

Line 57 - Especially, not specially.

Line 58 - both study and demonstrate are not needed

Line 62 - is grammatically incorrect and difficult to interpret

Line 64 - Remove promotes and recommends and consider replying with lends support.

Line 66 - This is a very long run-on sentence. Break up into multiple sentences

Line 67 - proposes not propose

Line 71 - is this legislation or recommendations?  To me there is a big difference between the law and a recommendation

Line 72 - remove true

Line 78 - I know what this sentence is saying, but it needs to be reworded to clearly express this view

Line 82 - Should be species not specie

Line 90 - remove applied and replace with more descriptive term mailed, emailed, given etc.

Line 94 - in a workshop not of a workshop. I’d also like to see more information about this workshop. How were Ss invited and what were told about the workshop?

Line 102 - information about not of

Line 104 - what does type of technical advice mean? Also, does flock size include breeding ewes only, all sheep etc.?

Line 108 - If identification means ear tagging please say ear tagging.

Line 116 - Section include no information about how these responses were coded

Line 125 - what is foot paring?

Line 131 - what is NRS?

Line 153 - The table does not include all the socio-demographics mentioned earlier. Please address why this is the case.

Line 168 - You need to include some information in Materials and Methods about how you coded and all the open-ended responses you included

Line 176 - what does “short” mean?  I thought you had them mark where they dock on visual analogue scale. Can't you report the mean length

Line 192 - This is unclear.

Line 204 - Does the n = 125 mean you only counted surveys that answered all these questions? You didn’t have an Ss that didn’t answer one or more of the painfulness scales?

Line 254 - Please make it clear what data you are basing these comparisons on and whether it is a far comparison.  You might consider discussing how Chilean sheep farmers differ from the general Chilean pop. if this info. is available.

Line 272 - I suppose it depends on what you think is the earliest possible age? Maybe the farmers think this is the earliest possible age?

Line 280 - what does performed accordingly mean?

Line 283 - most painful method of what? both ear notching and tail docking??

Line 288 - please mention time course and how this makes your statement far too simplistic

Line 300 - change to may be more difficult to justify

Line 309 - change to should be maintained

Line 310 - legislation is mandated by definition unless I am missing something

Line 328 - I have no way of checking whether the data support this claim

Line 351 - what scientific evidence is there that shows the pains caused are equivalent?

Line 363 - not clear why seeing blood explains this

Line 365 - it is only contradictory if you think that farmers make management decisions sole on the basis of pain experienced by animals - which they do not (nor do non-farmers considering pain)

Line 383 - why would a lack of formal education explain this? You need to explain why having a college degree makes your judgments of animal pain more accurate

Line 406 - need to explain what constitutes early

Line 412 - isn’t water and food more important?

Line 417 - it is also possible that they think it is painful, but dont care enough to do anything about it.

Author Response

We appreciate the reviewer’s comments and suggestions. We have considered almost all these recommendations in order to improve our paper. Hereby we enclose the survey as you requested.

Response to Reviewer 2 Comments

Line 14 - This sentence is very long. Please reword to be more precise

Action: Modified according to reviewer comment, see lines 14 to 18, two sentences now

Line 18 - Please mention how the farmers were recruited

Action: Modified according to reviewer comment, lines 18-19

Line 22 - Grammar: practices not practice. Should be farmer’s not farmers

Action: Modified according to reviewer comment, line 22

Line 23 - I fail to see how the authors judgment that painful practices have negative impact on welfare is relevant to this study. I would also avoid the subsequent judgment that farmers don’t do things “properly”. Properly according to whom?

Action: This paragraph was changed by: In general, husbandry practices are carried out without using analgesics and with painful methods despite to the agreement among farmers in the recognition of pain associated with these procedures, now lines 24-26. 

Line 43 - ungrammatical. Also please consider using the ISAP definition which stipulates that tissue damage is neither necessary or sufficient for pain.

Action: Modified according to reviewer comment, lines 44-45. We added the IASP reference

Line 45 - why do you define pain twice?  What is the point? 

Action: The second definition was deleted

Line 48 - diseases is clear, but conditions is very vague consider removing or clarifying

Action: We deleted the word conditions, line 49

Line 52 - It is not clear to me what this line is saying.  Researchers disagree about what? I doubt that they disagree that these procedures are painful, but rather about what ought to be done about the pain or how important it is.

Action: We clarified this sentence. Researchers disagree about the actual need to carry out some of these procedures (eg. tail docking and castration) in a routine way, without a scientific purpose. This sentence was modified, lines 52-55

Line 57 - Especially, not specially. 

Action: Modified according to reviewer comment, line 56

Line 58 - both study and demonstrate are not needed

Action: Modified according to reviewer comment, line 57

Line 62 - is grammatically incorrect and difficult to interpret

Action: We modified this sentence. Replaced by: These behaviors do not necessarily have beneficial effects for the animal or contribute to pain relief [3]. Lines 60-61

Line 64 - Remove promotes and recommends and consider replying with lends support.

Action: This sentence was modified: Scientific evidence has shown improvements on animal welfare using analgesics during painful procedures. Lines 62-63

Line 66 - This is a very long run-on sentence. Break up into multiple sentences

Action: This sentence was modified: Similarly, Chilean legislation [19] requires mitigation of animal suffering during painful husbandry practices and proposes four possible ways to reduce pain and improve animal welfare: 1) replacing the current husbandry practice by another non-surgical procedure that has demonstrated to improve animal welfare; 2) carrying out the husbandry practices at the earliest possible age; 3) use analgesics during these procedures; or 4) genetic selection, selecting for animals that do not present the feature that requires the husbandry procedure. Lines 64-70

Line 67 - proposes not propose

Action: Modified according to reviewer comment, line 66

Line 71 - is this legislation or recommendations?  To me there is a big difference between the law and a recommendation

Action: It has been replaced by: requirements instead of recommendations. Line 70

Line 72 - remove true

Action: Modified according to reviewer comment, line 71

Line 78 - I know what this sentence is saying, but it needs to be reworded to clearly express this view

Action: Modified according to reviewer comment: Pain evaluation and recognition in lambs is a complex process, mainly due to the fact that active pain behavior in these animals is not as intense as in other species [4]. Moreover, pain recognition may be influenced by the incorrect assumption that younger animals have an immature central nervous system [23,24]. Lines 77 to 80.

Line 82 - Should be species not specie

Action: Modified according to reviewer comment, line 81

Line 90 - remove applied and replace with more descriptive term mailed, emailed, given etc.

Action: Modified according to reviewer comment. It was replaced by ‘’given to participants to be completed”, line 95-96

Line 94 - in a workshop not of a workshop. I’d also like to see more information about this workshop. How were Ss invited and what were told about the workshop?

Action: As expressed in lines 92 to 96 sheep farmers were invited through the governmental and private institutions used to regularly send information and give technical advice to them; the places where they gathered for the workshop in each region were facilitated by these institutions.

Line 102 - information about not of

Action: Modified according to reviewer comment, line 103

Line 104 - what does type of technical advice mean? Also, does flock size include breeding ewes only, all sheep etc.?

Action: Technical advice means private or public advice on nutrition, reproduction or animal health topics. Flock size includes all sheep. We have included this information: The survey consisted of three sections. In the first section, sociodemographic information about the sheep farmers was obtained, including gender, age, region, educational level, as well as information related with their time experience as sheep farmer, farm size, flock size (all sheep) and type of technical advice (nutritional, reproductive, health or other) if it existed. Lines 103 to 106.

Line 108 - If identification means ear tagging please say ear tagging.

Action: Animal identification is used in general regardless of the method used (answer was open and could be ear tagging, ear notching or other), line 109

Line 116 - Section include no information about how these responses were coded

Action: We have included this information: The third section aimed to establish the farmer’s knowledge concerning animal welfare including individual perception and importance. Farmers were asked about had previously heard about animal welfare (yes/no), and also were asked to select from a multiple-choice question based on the five freedoms of animal welfare, and what animal welfare means for them. Besides, participants were asked to select one or more options from a pre-defined list the importance of having good animal welfare in their farms, considering the following alternatives: affects animal health and production; that it influences meat quality; that it is a consumer requirement; or animal welfare is indifferent to me. Also, they were asked to select and rank in order of importance the three main animal welfare issues in their farms from a pre-defined list: predation, lamb mortality, rounding up/transportation, husbandry practices, slaughter, facilities, nutrition/water availability and diseases. Lines 119-126.

Line 125 - what is foot paring?

Action: It is a frequently used synonym for hoof trimming, however we replaced it by hoof trimming. Line 133.

Line 131 - what is NRS?

Action: Numerical rating scale. Line 134

Line 153 - The table does not include all the socio-demographics mentioned earlier. Please address why this is the case.

Action: This is to avoid repeating information, because part of the sociodemographic information was already reported in the previous paragraph (text). Lines 157 to 163: The majority of sheep farmers were males (n = 145; 87.9%), 18 (10.9%) were females and only two persons did not answer this question (1.2%). The mean age of the participants was 47.3 ± 14.5 years, the mean time experience as sheep farmer was 23.5 ± 19.9 years. More than half of the farms had technical advice, either from the government (63/165; 38.2%) or private (41/165; 24.8%). In addition, most of the farms (115/165; 69.7%) were visited by a veterinarian, at least once a year. Table 1 shows further information about age of farmers, their educational level, number of ewes per farmer (flock size) and farm size (ha).

Line 168 - You need to include some information in Materials and Methods about how you coded and all the open-ended responses you included

Action: We have added this information: Sociodemographic information and open-ended responses from the second and third sections (routine husbandry procedures and general animal welfare perception) were coded as frequencies and percentages. The scores obtained in the NRS for each husbandry practice, were summarized in a frequency distribution table with their mode, median and range. Lines 138 to 142.

Line 176 - what does “short” mean?  I thought you had them mark where they dock on visual analogue scale. Can't you report the mean length

Action: This information is given in Material and Methods (Routine husbandry practices: lines 112-117): Additionally, for tail docking a diagram was used to mark the length at which the tail was docked and to justify the answer. The diagram was adapted from Fisher et al. [8] and farmers had to mark one of three options for tail length: Short = docked leaving little tail that does not cover the tip of the vulva in ewe lambs and at similar length in male lambs; Medium = tail stump covers the tip of the vulva in ewe lambs and at similar length in male lambs; Long = docked longer than Medium.

Line 192 - This is unclear.

Action: The vast majority of farmers selected the five alternatives from a pre-defined list. We have included this information in Material and Methods: Farmers were asked if they had previously heard about animal welfare (yes/no), and to select from a multiple-choice question based on the five freedoms of animal welfare [6] what animal welfare means to them. Participants were also asked to select one or more options from a pre-defined list regarding the importance of having good animal welfare in their farms, considering the following alternatives: it affects animal health and production; it influences meat quality; it is a consumer requirement; or animal welfare is indifferent to me. Lines 120 to 126.

Line 204 - Does the n = 125 mean you only counted surveys that answered all these questions? You didn’t have an Ss that didn’t answer one or more of the painfulness scales?

Action: We only included data from farmers that completed all the numerical rating scale (seven husbandry practices). There were some farmers that didn’t answer one or more husbandry practices of the numerical rating scale. See lines 217-218.

Line 254 - Please make it clear what data you are basing these comparisons on and whether it is a far comparison.  You might consider discussing how Chilean sheep farmers differ from the general Chilean pop. if this info. is available.

Action: We have modified this sentence as following: Sociodemographic features of sheep farmers (Table 1) are similar to the current national situation, in which the majority of farmers are males [25]. Lines 277-278.

Line 272 - I suppose it depends on what you think is the earliest possible age? Maybe the farmers think this is the earliest possible age?

Action: We agree with the reviewer comment. We modified this sentence: Current European and Chilean legislations [9,10,19], mandate performing painful husbandry practices in young animals. Lines 293-294.

Line 280 - what does performed accordingly mean?

Action: We have modified this sentence: Moreover, tail docking at 45 days, may induce long-term consequences, such as primary hyperalgesia and chronic pain [33]. Line 301-303.

Line 283 - most painful method of what? both ear notching and tail docking??

Action: For both ear notching and tail docking, line 306.

Line 288 - please mention time course and how this makes your statement far too simplistic

Action: We do not understand this comment, but we thought that reviewer refers to the word “quickness”. We modified this sentence adding the word “relative”. Line 310

Line 300 - change to may be more difficult to justify

Action: Modified according to reviewer comment, line 322.

Line 309 - change to should be maintained

Action: Modified according to reviewer comment, line 331.

Line 310 - legislation is mandated by definition unless I am missing something

Action: Modified according to reviewer comment, line 332

Line 328 - I have no way of checking whether the data support this claim

Action: We have clarified this information in Material and Methods (lines 120 to 126): Farmers were asked if they had previously heard about animal welfare (yes/no), and to select from a multiple-choice question based on the five freedoms of animal welfare [6] what animal welfare means to them. Participants were also asked to select one or more options from a pre-defined list regarding the importance of having good animal welfare in their farms, considering the following alternatives: it affects animal health and production; it influences meat quality; it is a consumer requirement; or animal welfare is indifferent to me.

Line 351 - what scientific evidence is there that shows the pains caused are equivalent?

Action: We have modified this sentence (lines 371 to 373): Castration was perceived by sheep farmers as the most painful husbandry practice for lambs, followed by tail docking and animal identification (Table 2), which is in agreement with scientific evidence [6,27,47,50]. - Scientific evidence has shown that castration is more painful than tail docking and identification. These findings are similar to those reported in the present study (farmer’s perception).

Line 363 - not clear why seeing blood explains this

Action: We agree with the reviewer comment. This was only a presumption, we deleted it

Line 365 - it is only contradictory if you think that farmers make management decisions sole on the basis of pain experienced by animals - which they do not (nor do non-farmers considering pain)

Action: We agree with the reviewer comment. There are many reasons of why farmers perform husbandry practices. We changed the word contradictory to “does not agree”. Line 386.

Line 383 - why would a lack of formal education explain this? You need to explain why having a college degree makes your judgments of animal pain more accurate…..

Action: There is one study in cattle that did not find an educational effect on pain perception (see Kielland et al. 2010: Dairy farmer attitudes and empathy toward animals are associated with animal welfare indicators). We modified redaction as following: According to the results of our study, farmers who had higher educational level had a lower pain perception; these farmers also had larger flock sizes. Having larger flocks may reduce the human:animal interaction and also pain perception because they do not see and handle their animals frequently; on the contrary farmers with smaller flocks may have a closer human-animal bond and hence a higher perception of pain in their animals because they round them up every day and have greater chances of watching them closer. Lines 405 to 410.

Line 406 - need to explain what constitutes early

Action: We modified this sentence as following: Animal identification, tail docking and castration are painful husbandry practices that are carried out by Chilean sheep farmers at later ages than recommended by international literature and using methods that may have a negative impact on animal welfare. Lines 431-433.

Line 412 - isn’t water and food more important? 

Action: We agree with the reviewer comment. It is more important and that was the farmer’s perception.

Line 417 - it is also possible that they think it is painful, but dont care enough to do anything about it.

Action: We agree with the reviewer comment, it is also possible. However considering the findings of our study we cannot assume that statement as a result or conclusion of our results. 

Round 2

Reviewer 2 Report

Ln 13 should read “simultaneously subjected”

Ln 20 remove “asked”

Ln 21 remove “by farmers” Still not clear what “there was agreement” refers to

Ln 24 id prefer to see associated or related rather than “affected” since it implies causation/experimental designs to many

Ln 25 remove “and with painful methods”

ln 33 please modify procedures to be more descriptive e.f painful procedures or common management practices

ln 35 again please change affected

ln 36 farm and flock size seems redundant

ln 37 remove duplicated word

ln 39 what is an early age? I’d drop the “which may..” part. Please go through entire MS and try to speak either about pain or the much more general term animal welfare to avoid repeating them both together.

I would reorganize the introduction so that it flows much better. Right now it jumps around too much.

Ln 60 Remove “these behaviours”

Ln 52 Please just say analgesics reduce pain.

Ln 70 if legislation requires it then why are all the sheep farmers not using it without penalty?

Ln 74 "In addition to these limitations…” Delete “an important problem…”

Ln 80 “As prey animals, ruminants…” tend to be more stoic.

Ln 83 Remove “in order to further…” Everyone thinks alleviating pain will improve welfare so no need to say it.

Ln 95 Please provide the title of the workshop as it was advertised to farmers. Consider including a sentence or two with how it was described to farmers in the adverts.

Ln 97 165 completed all questions in the survey? Please list how many took the survey AND how many took it and completed all questions.

Ln 99 move ethics approval to first line of materials and methods
Ln 105 how was farm size assessed (hectares, income, employees etc.) It is still not clear what type of technical advice means? Type of advice they’d like to receive more of…?

Ln 108 remove “related” Please be specific about exactly what these asked.

Ln 110 “they were asked to indicate the age…”

Please provide the exact question (translated)  in quotes

Ln 121 it is not clear if they could select more than one response options regrind the five freedoms

I strongly recommend the authors create a table that include each variable they assessed and how it was coded to make it easy to see ho they coded response options. Section 2.4 may necessitate its own table.

ln 140 how were they coded?

Ln 145 replace effect with association or relationship

Ln 146 do you provide information on how variables were coded for regression?

Ln 147 Have the authors considered creating an aggregate pain score for all measures and modelling this?

Ln 160 I would call this “source of technical advice” again depending on how the question was asked

Ln 177 Please provide info on how these open ended responses were coded

Ln 181 remove “A” - same on Ln 177

Ln 194 remove A

Ln 199 what were the other response options besides farm personal or themselves?

Ln 202 They associated welfare with all of the 5 freedoms? Please remove associated and use the term that reflects what they were actually asked to do.

Ln 205 remove A

Ln 207 Associated with the fact

Ln 224 I do not think it is necessary to give raw counts for each of the numbers on the NRS. Simply reporting the means is sufficient.

Ln 243 Except for shearing? As stated might be interesting to create an aggregate pain score and see what sociodemographic predict it.

Ln 296 “methods…" that are less painful

Ln 298 is this true? Do 2 studies justify the very general claim made here? I am doubtful and would suggest revising to be more cautious.  e.g. “Some studies suggest rubber ring…”

Ln 303 questionable. We are not aware of any studies that…

Ln 326 replace totality with nearly all farmers

Ln 340 farmers agreed that animal welfare

Ln 346 However, animal welfare in extensive… is affected by…

Ln 359 The author suggests there is scientific evidence comparing castration to tail docking and ear notching. I do not think this is so. Please explain

Ln 365 I still don’t find this part persuasive. Why would education

Ln 388 please remove 'dirty work employees’ and re-phrase. This is not clear. You may also consider referencing other work involving farmer perceptions of painful procedures in other species

Ln 393 or it could be some other difference between them that you didn’t measure

Ln 417 Id like to see a very brief mention of the limitations of this study. For one, self section bias in that the farmers that showed up may be the most interested in animal welfare.

Ln 419 Please mention the fact that analgesia is rarely used.

Author Response

We appreciate the reviewer’s comments and suggestions. We have considered almost all these recommendations in order to improve our paper.

Response to Reviewer 2 Comments

Ln 13 should read “simultaneously subjected”

Action: Modified according to reviewer comment, line 13.

Ln 20 remove “asked”
Action: We deleted the word “asked”

Ln 21 remove “by farmers” Still not clear what “there was agreement” refers to

Action: We deleted “by farmers”.  We modified the sentence, lines 20 to 22: Castration and tail docking were perceived as the most painful practices and farmers agreed among them that these routine husbandry practices cause severe pain to animals.

Ln 24 id prefer to see associated or related rather than “affected” since it implies causation/experimental designs to many
Action: We replaced the word “affected” by ´´were associated with´´.

Ln 25 remove “and with painful methods”

Action: We deleted ‘’and with painful methods”.

ln 33 please modify procedures to be more descriptive e.f painful procedures or common management practices

Action: This word was replaced by “common husbandry practices”, line 33.

ln 35 again please change affected

Action: Modified according to reviewer comment, line 35.

ln 36 farm and flock size seems redundant

Action: These terms are different variables, although they are related because usually large farms have large flocks and viceversa but we used them as separated variables. It was further explained in material and methods.

ln 37 remove duplicated word

Action: We deleted the duplicated word ‘’agreement’’

ln 39 what is an early age? I’d drop the “which may..” part. Please go through entire MS and try to speak either about pain or the much more general term animal welfare to avoid repeating them both together.

Action: This sentence was modified according to reviewer comment. Also, “Early age” was replaced by “young animals”. Line 39.

I would reorganize the introduction so that it flows much better. Right now it jumps around too much.

Action: Introduction was reorganized according to reviewer suggestions.

Ln 60 Remove “these behaviours”

Action: This sentence was deleted

Ln 52 Please just say analgesics reduce pain.

Action: This sentence was modified: Scientific evidence has shown that analgesics reduce pain during husbandry procedures [6,16,17]. Line 60.

Ln 70 if legislation requires it then why are all the sheep farmers not using it without penalty?

Action:  Chilean legislation requires mitigation of animal suffering during painful husbandry practices. However, Chilean legislation only proposes (but not requires) ways to reduce pain and improve animal welfare, and one of them is use analgesics during husbandry procedures. Lines 61 to 68.

Ln 74 "In addition to these limitations…” Delete “an important problem…”

Action: This sentence was modified according to reviewer comment.

Ln 80 “As prey animals, ruminants…” tend to be more stoic.

Action: This sentence was modified according to reviewer comment.

Ln 83 Remove “in order to further…” Everyone thinks alleviating pain will improve welfare so no need to say it.

Action: This sentence was modified according to reviewer comment. Lines 71 to 73.

Ln 95 Please provide the title of the workshop as it was advertised to farmers. Consider including a sentence or two with how it was described to farmers in the adverts.

Action: The title of the workshop was: Improving handling practices on sheep farms. This sentence was added, line 92.

Ln 97 165 completed all questions in the survey? Please list how many took the survey AND how many took it and completed all questions.

Action: This sentence was modified according to reviewer comment, line 95 to 98:  Of a total of 747 existing farmers in the mentioned regions [25], 180 attended the workshops and took the survey and 165 completed all questions. However, in the case of the numerical rating scale (NRS) for pain perception of routine husbandry practices only 125 farmers rated all husbandry practices.

Ln 99 move ethics approval to first line of materials and methods

Action: This sentence was moved to first line, lines 85 to 86.

Ln 105 how was farm size assessed (hectares, income, employees etc.) It is still not clear what type of technical advice means? Type of advice they’d like to receive more of…?

Action: Farm size was asked in hectares and type of technical advice was asked in terms of areas (nutritional, reproductive, health or none of them). This sentence was modified: The survey consisted of three sections. In the first section, open-ended questions about farmer’s sociodemographic information was obtained, including gender, age, region, educational level, as well as information related with their time experience as sheep farmer (years), farm size (hectares), flock size (all sheep). Also, farmers were asked to select one or more options regarding if their farm had technical advice in the following areas: nutritional, reproductive, health, none of them. Lines 102 to 106.

Ln 108 remove “related” Please be specific about exactly what these asked. 

Action: The word “related” was removed. These specifications are mentioned in lines 108 to 109: The second section of the survey, contained open-ended questions of how sheep farmers carry out the following husbandry practices in lambs:

Ln 110 “they were asked to indicate the age…” 

Please provide the exact question (translated)  in quotes

Action: This sentence was modified according to reviewer comment. Lines 110 to 111: For each procedure, they were asked by open question if they carry out each practice, and if they do, “at what age they carry it out”.

Ln 121 it is not clear if they could select more than one response options regrind the five freedoms

Action: Yes, they could select more than one response. This sentence was modified, lines 120 to 124: Farmers were asked if they had previously heard about animal welfare (yes/no), and to choose what animal welfare means to them by selecting one or more of the following options: animals do not suffer from hunger and thirst; animals do not suffer from pain, injury and diseases; animals do not suffer from discomfort; animals do not suffer from fear and distress; animals can express normal behavior.

I strongly recommend the authors create a table that include each variable they assessed and how it was coded to make it easy to see ho they coded response options. Section 2.4 may necessitate its own table.

Action: We feel that the table is not necessary because we included almost all this information as text in material and methods. For section 2.4 we included “(as table 2 of the results section)” in order not to repeat the same table without frequencies, line 134.

ln 140 how were they coded?

Action: The scores obtained in the NRS for each husbandry practice were coded as counts and frequencies.

Ln 145 replace effect with association or relationship

Action: We replaced by association.

Ln 146 do you provide information on how variables were coded for regression?

Action: We added this information and modified the following sentence (lines 148 to 150):  Linear regression models with Poisson distribution were fitted in order to assess the effect of the continuous variables (age, time experience as sheep farmer) and factors (gender, educational level, flock size, farm size) on the pain scores associated to each husbandry practice.

Ln 147 Have the authors considered creating an aggregate pain score for all measures and modelling this?

Action: We did it and we found similar results. However, we considered more accurate to estimate the association between sociodemographic factors and pain perception associated to each husbandry practice, due to the high dispersion of pain scores for some of them. We used an aggregate pain score for Spearman correlations, lines 144 to 146: Spearman correlations were made between pain scores (median pain scores) that sheep farmers perceived associated to husbandry practices.

Ln 160 I would call this “source of technical advice” again depending on how the question was asked

Action: This comment was clarified on lines 102 to 106: Also, farmers were asked to select one or more options regarding if their farm had technical advice in the following areas: nutritional, reproductive, health, none of them.

Ln 177 Please provide info on how these open ended responses were coded
Action: This information is provided on lines 140 to 142:
Sociodemographic information and open-ended responses from the second and third sections (routine husbandry procedures and general animal welfare perception) were coded as frequencies and percentages.

Ln 181 remove “A” - same on Ln 177
Action: The word “A” was deleted

Ln 194 remove A
Action: The word “A” was deleted

Ln 199 what were the other response options besides farm personal or themselves?

Action: As mentioned in lines 111 to 112, this was an open-ended question and was coded as frequency and percentage of response. There were no other responses anyway

Ln 202 They associated welfare with all of the 5 freedoms? Please remove associated and use the term that reflects what they were actually asked to do.

Action: Yes, they associated welfare with all of the five freedoms. We removed associated and modified this sentence, lines 204 to 206: Regarding animal welfare [29], 83.6% of participants selected it all the options given as explained in subsection 2.3, meaning that animals do not suffer from thirst, hunger and malnutrition, pain, injuries, diseases, discomfort, fear and distress, and the possibility to express normal behaviors.

Ln 205 remove A

Action: The word “A” was deleted

Ln 207 Associated with the fact
Action: This sentence was modified according to reviewer comment.

Ln 224 I do not think it is necessary to give raw counts for each of the numbers on the NRS. Simply reporting the means is sufficient. 

Action: We agreed with the reviewer comment, raw counts were deleted.

Ln 243 Except for shearing? As stated might be interesting to create an aggregate pain score and see what sociodemographic predict it.

Action: We responded to this comment previously. We did it and we found similar results

Ln 296 “methods…" that are less painful

Action: This sentence was modified according to reviewer comment, line 308.

Ln 298 is this true? Do 2 studies justify the very general claim made here? I am doubtful and would suggest revising to be more cautious.  e.g. “Some studies suggest rubber ring…”

Action: We added one more reference [6] and modified this sentence according with to reviewer comment, lines 309 to 311: However some studies suggest that rubber ring castration causes greater pain associated behavioral changes [6,23,37].

Ln 303 questionable. We are not aware of any studies that…

Action: This sentence was modified according to reviewer comment, line 315.

Ln 326 replace totality with nearly all farmers

Action: This sentence was modified according to reviewer comment, line 338.

Ln 340 farmers agreed that animal welfare
Action: This sentence was modified according to reviewer comment, line 352.

Ln 346 However, animal welfare in extensive… is affected by…

Action: This sentence was modified according to reviewer comment, lines 358 to 359.

Ln 359 The author suggests there is scientific evidence comparing castration to tail docking and ear notching. I do not think this is so. Please explain

Action: We suggested there is scientific evidence comparing castration, tail docking and identification (regardless of the method used). There are three articles that have compared husbandry practices and have reported similar results to our study. The study of Tamioso et al. (2017): “Attitudes of South Brazilian sheep farmers to animal welfare and sentience”, which compared farmer’s perception to these painful practices and reported that castration was mostly associated to cause “severe and very severe level of suffering”, following to tail docking “moderate suffering” and identification “mild and moderate suffering”. The study of Scott et al. (2003) “Evaluation of welfare state based on interpretation of multiple indices” which reported that castration was perceived by veterinarians as a more painful practice than tail docking. Also, the study of Grant et al. (2004): “Behavioural responses of lambs to common painful husbandry procedures” which compared the incidence of active pain behaviours and abnormal postures associated to commonly used husbandry procedures. We deleted one reference ‘’[50]’’.

Ln 365 I still don’t find this part persuasive. Why would education 

Action: This sentence was deleted.

Ln 388 please remove 'dirty work employees’ and re-phrase. This is not clear. You may also consider referencing other work involving farmer perceptions of painful procedures in other species
Action:
This sentence was suggested by Reviewer 1. However, we modified this sentence: Managers of larger flocks may be distanced from participating in painful routine procedures, lines 397 to 398.

Ln 393 or it could be some other difference between them that you didn’t measure

Action: We agreed with the reviewer comment. There are many factors that may affect farmer’s perception eg. their attitudes; but for this we would need to do an observational study in each farm and could also be a limitation of our study.

Ln 417 Id like to see a very brief mention of the limitations of this study. For one, self section bias in that the farmers that showed up may be the most interested in animal welfare.

Action: We agreed with the reviewer comment. We added the following sentence: This study may be limited by the selection of farmers that attended an industry-government continuing education event and a self-administered survey format. This process may have selected from a cohort of more generally literate farmers than the population targeted. This is the first study in Chile that approaches how sheep husbandry practices are performed and farmer’s perception of pain and animal welfare; more on farm studies should follow. Lines 427 to 431.

Ln 419 Please mention the fact that analgesia is rarely used.

Action: This sentence was modified according to reviewer comment, line 435.

Animals EISSN 2076-2615 Published by MDPI AG, Basel, Switzerland RSS E-Mail Table of Contents Alert
Back to Top